

# Automated iceberg detection using Landsat: method and example application in Disko Bay, west Greenland

Jessica Scheick[1,2], Ellyn M. Enderlin[1,2], and Gordon Hamilton[a,b,†]

[1]School of Earth and Climate Sciences, University of Maine, Orono, ME, USA
[2]Climate Change Institute, University of Maine, Orono, ME, USA
[a]formerly at: School of Earth and Climate Sciences, University of Maine, Orono, ME, USA
[b]formerly at: Climate Change Institute, University of Maine, Orono, ME, USA
[†]deceased

**Correspondence:** Jessica Scheick (jbscheick@gmail.com)

**Abstract.** Over the last two decades, the flux of icebergs into Greenland's fjords and coastal waters has increased, concurrent with changes in mass loss and dynamics of Greenland's marine-terminating outlet glaciers. Icebergs impact fjord circulation and stratification, freshwater flux, and ecosystem structure and pose a hazard to marine navigation and infrastructure, yet they remain a relatively understudied component of the ice–ocean system. Icebergs are easily detected in optical satellite imagery,

but manual analysis to derive an iceberg size distribution time series is time prohibitive and partially cloudy scenes pose a challenge to automated analysis. Here we present a novel, computationally simple machine learning-based cloud mask for Landsat 7 and 8. This mask is incorporated into a larger iceberg delineation algorithm that allows us to extract iceberg size distributions, including outlines of individual icebergs, for cloud-free and partially cloud-covered Landsat scenes. We applied the algorithm to the Landsat archive covering Disko Bay, west Greenland, to derive a time series of iceberg size distributions

from 2000–2002 and 2013–2015. The time series captures the seasonal signal in ice cover resulting from the annual cycles of sea ice formation and breakup and calving of Jakobshavn Isbræ, the dominant source of icebergs in Disko Bay. We note a change in this annual signal during the latter time period, likely a direct result of changes in the calving regime of Jakobshavn Isbræ. During 2000–2002, Jakobshavn Isbræ's floating ice tongue disintegrated and disappeared, transitioning the glacier from low energy, tabular iceberg calving to high energy, full thickness calving and the production of more small icebergs. This

transition is also evident in the change in the number of small (~225 m$^2$) icebergs found on the bay, which increased during the latter period. The change in the number of small icebergs also led to increasingly negative power law slopes fit to the iceberg size distribution time series. This work suggests that iceberg size distribution time series may provide useful insights into changes in calving dynamics and the physics of iceberg decay, while aiding marine and navigation safety in iceberg laden waters.

# 1 Introduction

Over the last two decades, as Greenland's marine-terminating outlet glaciers have retreated, thinned, and accelerated, the flux of icebergs to Greenland's fjords and coastal waterways has increased. Until recently, these icebergs remained a relatively little



studied phenomena, despite their physical, ecological, and socioeconomic importance. Unlike point sources of freshwater such as meltwater streams, rivers, and subglacial conduits, icebergs represent a distributed source of freshwater to the ocean (Silva et al., 2006). Depending on where drifting icebergs melt, their freshwater may play an important role in fjord stratification (Sutherland et al., 2014), overturning circulation (Bamber et al., 2012), and/or ecosystem structure (Greene et al., 2008). The physical characteristics of icebergs can provide important insights into the dynamics of the glaciers that produced them (Bassis and Jacobs, 2013), and the number and physical dimensions of icebergs in a fjord or shelf area places a constraint on their potential freshwater flux (Sutherland et al., 2014; Enderlin et al., 2016). Recent analyses suggest that time series of iceberg melting can be used to infer variations in subsurface ocean conditions and ice–ocean interactions near glacier termini (Moon et al., 2017; Enderlin et al., 2018). Icebergs also pose hazards to marine navigation and offshore infrastructure in the Arctic, where natural resource exploration and shipping have recently expanded (Pizzolato et al., 2014).

Arguably the most important location to study icebergs around Greenland is along the central west coast (Fig. 1). As these icebergs transit through the Labrador Sea into the shipping lanes of the North Atlantic (e.g., Bigg et al., 1997), their meltwater has the potential to disrupt the formation of North Atlantic Deep Water (NADW) (Fichefet et al., 2003). Icebergs in Disko Bay are primarily sourced from Jakobshavn Isbræ (Greenlandic: Sermeq Kujalleq), which is the most prolific producer of icebergs in Greenland (Enderlin et al., 2014). Icebergs calve into the deep Jakobshavn (Ilulissat) Isfjord and traverse the ~60 km long fjord until it empties into Disko Bay. At the mouth of the fjord, a large bedrock sill shallows the water from close to 800 m in the fjord to about 200 m at the entrance to the bay (Schumann et al., 2012; Gladish et al., 2015), creating a chokepoint where large (>200 m deep) icebergs become stranded. As such, the sill acts as an iceberg filter, restricting the size and number of icebergs entering Disko Bay.

North of the fjord mouth, along the eastern shore of the bay, sits the town of Ilulissat. Residents here depend heavily on fishing and tourism, so navigability of Disko Bay's iceberg laden waters is of critical importance to their livelihoods. Anecdotal evidence suggests that the number and size distribution of icebergs present in Disko Bay has changed in recent decades as Jakobshavn Isbræ has thinned and retreated. Specifically, marine navigators and fishermen noted that in recent years the bay is frequently covered with a large number of small icebergs, making navigation and fishing difficult. However, there has not been a comprehensive analysis of temporal variations in iceberg size distributions in Disko Bay, or elsewhere around Greenland, to date. A primary reason for the dearth of iceberg size distribution data is that there can be tens of thousands of icebergs in a fjord or bay at any given point in time (Enderlin et al., 2016). This large number of icebergs poses a challenge to manual iceberg identification using satellite images. Iceberg detection and size characterization is further hindered by the presence of clouds in visible to near infrared satellite imagery. Excluding images with partial cloud cover would considerably limit the temporal resolution of any derived dataset, resulting in potential aliasing of changes in iceberg size distributions over time. Therefore, efficient mapping of icebergs mandates that clouds are automatically distinguished from snow and ice in satellite images (i.e. automated cloud masking).

The primary challenge in mapping clouds is that clouds and snow/ice share similar spectral properties, including a large portion of the electromagnetic spectrum that is typically sensed by satellite platforms. Clouds and snow/ice are especially similar in the visible wavelengths (highly reflective) and thermal wavelengths (cold), properties that often allow clouds to be





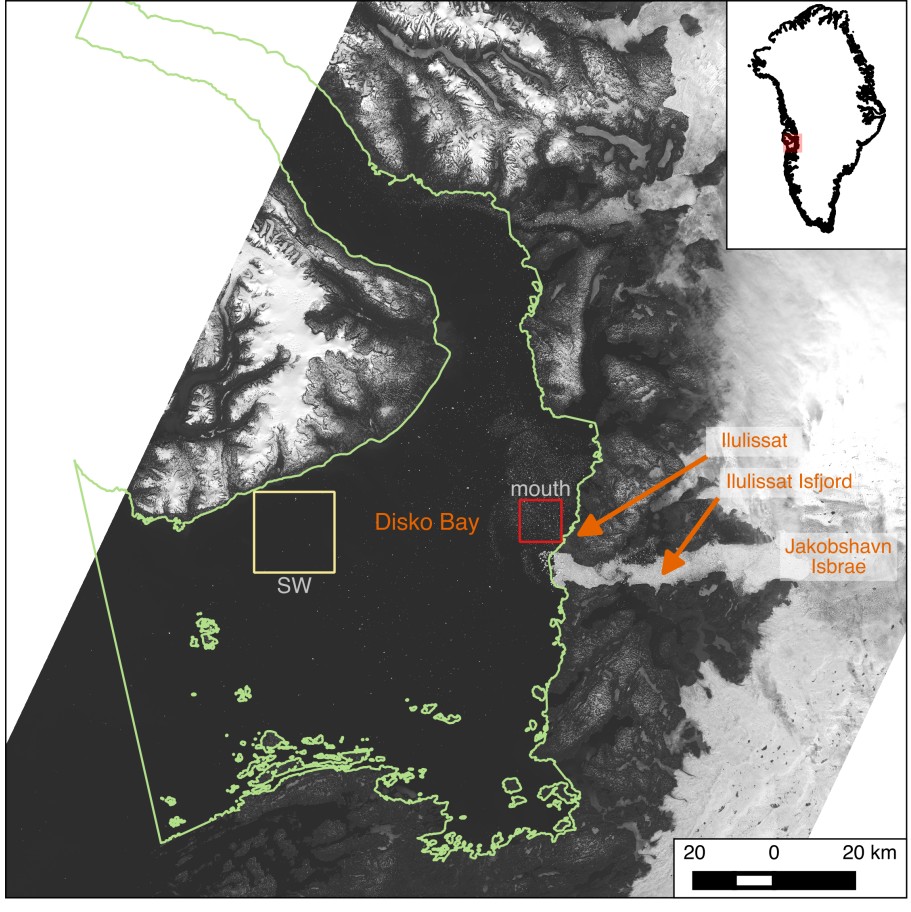

**Figure 1.** Map of Disko Bay, showing the location of the study region in west Greenland. Features of note include Jakobshavn Isbræ (i.e., the iceberg source), Ilulissat Isfjord, the town of Ilulissat, the study region of interest (ROI, green outline), and the regions in which icebergs were manually delineated (red (mouth) and yellow (southwest) outlines). Background is a Landsat 8 panchromatic image from 27 June 2015.

readily distinguished from relatively warm and dark-colored land surfaces. A number of different cloud detection schemes exist, including a few designed specifically for the Landsat sensors (e.g. Irish et al., 2006; Oreopoulos et al., 2011; Kustiyo et al., 2012; Zhu and Woodcock, 2012; Foga et al., 2017). Common approaches for detecting clouds in optical imagery include band thresholding, both of individual bands and combinations of bands, in order to exploit differences in the spectral properties of the different media (e.g. Racoviteanu et al., 2009). Often, several thresholding approaches are combined and/or advanced computing and image analysis techniques are utilized to achieve the best results (e.g. Hall et al., 1995; Riggs and Hall, 2002; Scaramuzza et al., 2012; Zhu and Woodcock, 2012). However, the existing cloud masking approaches often fail to accurately differentiate clouds from ice and snow surfaces (Oreopoulos et al., 2011; Foga et al., 2017), including or excluding both clouds and snow/ice simultaneously. Dozier (1989); Riggs and Hall (2002); Irish et al. (2006); Jedlovec (2009), and many others provide insight on various approaches to generating cloud masks for optical satellite images, including advanced image



analysis techniques based on machine learning (Lee et al., 1990; Scaramuzza et al., 2012; Hughes and Hayes, 2014) and edge detection (Zhu and Woodcock, 2012).

To address the need for an iceberg dataset that captures potentially important changes in iceberg size distributions over time, we have developed a Landsat-based automated iceberg delineation algorithm that incorporates a machine learning-based cloud mask. Here we present details of the algorithm and results from Disko Bay, west Greenland, to demonstrate the utility of the algorithm to explore temporal changes in iceberg size distributions commensurate with glacier change. Because icebergs pose a considerable hazard for marine navigation, the method we present in Sect. 2.1–2.3 was developed so that it can be implemented locally on a standard computer and in near real-time. We evaluate our algorithm's performance by means of qualitative and quantitative analyses of data from Disko Bay, including comparison to a manually derived validation dataset (Sect. 2.4–2.5). The results of the validation procedures and the iceberg size distribution time series produced for Disko Bay are presented in Sect. 3. Finally, we compare changes in iceberg size distributions between the 2000–2002 and 2013–2015 time periods to explore the use of iceberg size distribution time series as a means to infer changes in glacier dynamics (Sect. 4).

## 2  Methods

### 2.1  Location Selection

Due to the large number of icebergs present in Disko Bay (Fig. 1), the pronounced changes in iceberg calving style and volume from the primary iceberg source (Jakobshavn Isbræ) (e.g. Amundson et al., 2008; Joughin et al., 2008; Cassotto et al., 2015, and references therein), and the importance of icebergs to the local communities, we identified a pressing need for an automated approach to delineate icebergs in both cloud-free and partially cloud-covered Landsat imagery of this region. Automated optical approaches, such as that used below, rely on the contrast between bright iceberg surfaces and the comparatively dark water surrounding them. Thus, they are unable to distinguish individual icebergs that are tightly packed. To avoid problems associated with automated detection of icebergs surrounded by sea or brash ice or dammed by icebergs stranded on the sill at the mouth of Ilulissat Isfjord, we focused our study on the area seaward of the sill. The exclusion of the fjord mouth enabled us to avoid detections of erroneously large "icebergs" that were really a combination of multiple icebergs stranded close together.

### 2.2  Imagery Selection

Changes in the style, and likely size and volume, of icebergs calved from Jakobshavn Isbræ initiated in the late 1990s and continued throughout the 2000s as the glacier's terminus retreated (Amundson et al., 2008; Joughin et al., 2008). In order to capture these changes and test the utility of iceberg size distribution time series for quantifying dynamic change, we focus our method development and application on Landsat images. The MODIS constellation also covers our time period of interest, but it has relatively coarse spatial resolution (minimum 250 m for MODIS versus 15 m for Landsat), thereby limiting the minimum size of icebergs we can detect (minimum area of 225 $m^2$ for Landsat versus 62,500 $m^2$ for MODIS). Given our interest in changing iceberg size distributions and the relatively small iceberg sizes found in lulissat Isfjord (Enderlin et al., 2016),



we consider only the Landsat suite of sensors and focus on deriving an automated algorithm that works across the archive. The results presented herein are from the ETM+, OLI, and TIRS sensors; results from 2000–2002 were collected by Landsat 7 (ETM+) while results from 2013–2015 were collected by Landsat 8 (OLI and TIRS). Both Landsat 7 and 8 have a panchromatic band with a spatial resolution of 15 m, visible through short-wave infrared (SWIR) bands at 30 m spatial resolution, and thermal

bands with 60–100 m (Landsat 7–8, respectively) spatial resolution. For processing steps that simultaneously utilize multiple bands of different native resolutions, such as in the cloud masking step, scenes are upsampled to the highest resolution of any of the input bands. For example, the cloud mask is generated at 30 m pixel resolution, then each mask pixel is parsed into four 15 m by 15 m pixels that each have the same value as the parent pixel. The potential impacts of this resampling are discussed below.

Pre-collection Landsat scenes were downloaded from the USGS EarthExplorer website (earthexplorer.usgs.gov). We used Landsat scenes spanning path numbers 10–11 and row numbers 11–12 because they provide the most complete coverage of Disko Bay. Multiple scenes collected during the same pass were mosaicked to maximize coverage. The relatively high latitude of Disko Bay (~68.5–70 degrees North) provided the advantage of a satellite repeat interval better than the 16 day standard repeat time for Landsat with the concurrent disadvantage of limiting our analysis seasonally. To minimize inclusion

of springtime scenes that contained extensive sea ice (Cassotto et al., 2015), even during periods of sufficient solar illumination (February–October), only scenes with collection dates from May to October were used. Based on these parameters, the number of image acquisition dates per year ranged from 11–12 for Landsat 7 (2000–2002 period) and 16–24 for Landsat 8 (2013–2015 period). Although we developed a cloud masking procedure to enable the use of Landsat scenes containing partial cloud cover, we manually screened scenes to exclude those with such extensive cloud or sea ice cover that icebergs could not be manually

identified. Approximately 70% of the available images contained manually-identifiable icebergs, so that 9–14 (median 10) image swaths were used for a given year.

## 2.3 Algorithm

A schematic of the procedure to detect icebergs in both cloud-free and partially cloudy Landsat scenes is shown in Fig. 2 and described in detail below.

### 2.3.1 Cloud Masking

In order to accurately map icebergs, clouds must first be masked out of the Landsat scenes. We tested a number of different cloud masking techniques, ultimately deriving a novel, computationally simple, machine learning-based approach.

### 2.3.2 Ratio-based Cloud Mask

A number of band ratio and other threshold-based approaches to automatically discriminate snow/ice from clouds were tested;

however, only a few of the most common ones and their results are discussed in detail below. These examples capture the




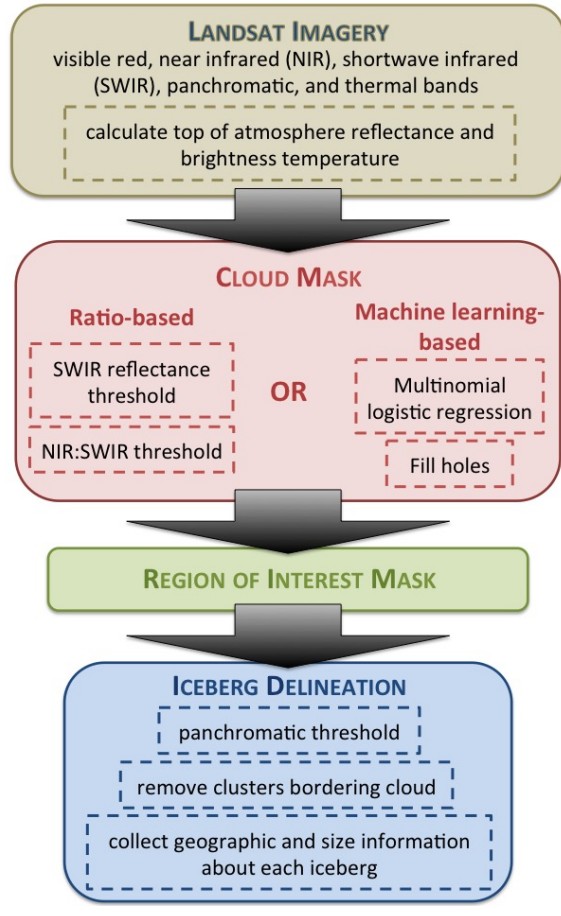

**Figure 2.** Schematic illustration of the steps of the automated iceberg delineation algorithm.

essence of the challenge in differentiating icebergs and clouds in optical satellite imagery, illustrating the problems encountered with all tested spectral-based methods.

Cloud identification is often accomplished by thresholding of normalized indices, ratios, and/or reflectance values that exploit the high reflectivity of clouds in certain wavelengths of the electromagnetic spectrum. One approach is to identify clouds simply

5   by applying a threshold to reflectance values in the SWIR part of the spectrum (centered at roughly 1.6 μm) (e.g. Warren, 1982; Dozier, 1989; Riggs and Hall, 2002). Another approach uses a ratio of the visible red and SWIR bands (wavelengths centered at roughly 0.66 and 1.6 μm, respectively) (Racoviteanu et al., 2009). In an effort to further exploit the spectral differences between clouds and snow/ice in these two bands, we computed a normalized index of the two bands (red-SWIR normalized index). We also tested a ratio of the near-infrared (NIR) to SWIR bands (centered at roughly 0.8 and 1.6 μm, respectively),

10   ultimately generating a ratio-based cloud mask using a combination of the NIR:SWIR ratio and SWIR reflectance.





### 2.3.3 Machine Learning-based Cloud Mask

In an effort to capitalize on the textural differences between clouds and icebergs, rather than relying solely on spectral differences, we tested methods that rely on edge detection to identify features or classify images into segments (i.e., segmentation). If successful, these methods could potentially enable the effective identification of icebergs regardless of cloud presence (i.e. only

icebergs would be detected), eliminating the need for a cloud masking step. Edge detection quantitatively identifies portions of an image where there are sharp transitions in brightness by comparing the value of a pixel to the values of the neighboring pixels. How many and which neighboring pixels are used is given as an input to the algorithm and the output image illustrates where the so-called edges occur. The derived edge maps can then be processed using feature detection or segmentation methods to identify different features. Feature detection uses the edge map to identify features with certain types of edge characteristics

(e.g. fuzzy, gradual edges of clouds or sharp edges of icebergs). Segmentation builds out from seeded regions (of unknown composition, unless they are manually identified) within the edges in the edge map.

Finally, we applied a machine learning technique called multinomial logistic regression to mask clouds in the available Landsat scenes. Multinomial logistic regression is a machine learning technique wherein a classified training set is provided as input and the computer generates a series of functions to relate the image pixel values and their classifications. Then,

the model is run on the validation pixel datasets and the computer predicted values are compared to the actual values. To use multinomial logistic regression to create a cloud mask, we first had to train the computer model. Groups of pixels were manually classified across six scenes with different cloud types. The pixel clusters were classified as either open water, opaque cloud, thin cloud with water underneath, thin cloud with iceberg underneath, or iceberg through clear sky. The top of atmosphere (TOA) reflectances and brightness temperatures were extracted for each classified pixel in the visible, NIR, SWIR, and thermal

wavelengths. These spectral signatures and associated classifications were combined to create training and validation datasets. Pixels from two scenes, one from each Landsat 7 and 8, were kept separate as a secondary validation dataset, while pixels from the remaining four scenes were randomly split into training and validation sets.

We trained and implemented our multinomial logistic regression model using the Python machine learning scikit-learn library (Pedregosa et al., 2011). A sensitivity analysis of our model allowed us to optimize the model for accuracy while

minimizing the number of required inputs and thus also minimizing computational requirements. The final model used only four spectral bands to classify pixels: TOA reflectance in the visible red, NIR, and SWIR bands (centered at roughly 0.66, 0.8, and 2.2 µm, respectively) and brightness temperature (centered at 11.5–12 µm). For each scene, a cloud mask was constructed from the model output and applied to the panchromatic band prior to iceberg identification. To improve the accuracy of the cloud mask over the central regions of very bright, white cumulus clouds where sensor saturation was common, groups of pixels

smaller than a ten by ten square structuring element (300 m by 300 m) classified by the model as ice but completely surrounded by pixels classified as clouds were added to the cloud mask. The rest of the classifications produced by the model were designated non-cloud. Although the model was also trained to classify ice pixels, we chose to identify icebergs as described below because visual inspection showed that thresholding tended to produce better results (e.g. it captured more of the icebergs) and it allowed us to optimize the performance of the model for identifying clouds. Prior to application, cloud masks were





morphologically opened (eroded and then dilated) using a cross shaped structuring element in order to remove small holes in the mask.

The red, NIR, SWIR, and thermal bands used to create the cloud mask have a lower spatial resolution (30–100 m pixels) than the panchromatic band (15 m pixels) to which the mask was applied. Thus, we ignored potential mixed pixel effects and

assumed that all higher-resolution pixels that fall within the cloud mask were clouds, resulting in a conservatively upsampled (i.e.,15 m resolution) cloud mask.

### 2.3.4 Land/Region of Interest Mask

To provide a consistent region for analysis across scenes, a region of interest (ROI) mask was applied subsequent to the generation and application of a cloud mask (Figs. 1 and 2). The ROI mask was generated by applying a 100 m buffer seaward

of the coastline and then manually connecting shorelines across the mouths of fjords and the bay (Fig. 1). The coastline shapefile used for land boundaries was provided by Asiaq (Asiaq, 2014). Application of the ROI mask excludes all pixels outside the ROI from thresholding during the iceberg delineation step.

### 2.3.5 Iceberg Delineation

Once land and clouds were masked out of the TOA reflectance of the panchromatic band, the masked images were used

to detect icebergs in the open water regions. Reflectance values in the panchromatic band greater than 0.19 were classified as iceberg. This threshold value was determined based on extensive qualitative inspection of different thresholds across ten scenes and spanning both Landsat sensors. The value was chosen to maximize the number of icebergs detected while limiting the number of false positive iceberg detections (i.e. identification as unmasked clouds or sea ice as "icebergs"). Given the stark contrast in brightness of icebergs from the surrounding water, the iceberg mapping results are fairly insensitive to the threshold

value. To quantify the influence of uncertainty in the threshold value, we ran the algorithm for one of our Landsat 8 scenes using higher and lower thresholds and compared the resulting iceberg delineations as well as the size distributions of icebergs to those derived using the 0.19 reflectance threshold. The results of this sensitivity analysis are presented in Sect. 2.5.

After thresholding, adjoining pixels were assumed to be part of the same iceberg and were clustered accordingly. For ice clusters that bordered the cloud mask, the number of water and cloud pixels bordering each cluster were counted, and those

clusters with a ratio of cloud to water pixels greater than 0.4 were reclassified as cloud edges. To avoid inclusion of portions of icebergs within fjord mouths or along scene boundaries, any iceberg polygon touching the ROI polygon or a scene edge was removed. The remaining clusters were classified as iceberg and outlined automatically by the computer, with each group of pixels stored as a unique iceberg polygon. The resulting shapefiles of iceberg polygons were used for the analysis of iceberg parameters presented below.



## 2.4 Iceberg Detection Screening

The accuracy and reliability of iceberg detections was determined through semi-automated methods. The time series of iceberg size distributions was inspected and scenes with an ice–open water ratio $>0.008$ or maximum iceberg area $>2\ \mathrm{km}^2$ were flagged as potential outliers, and the icebergs exceeding the latter size threshold were visually inspected. For all scenes processed by the algorithm, each iceberg shapefile was overlaid on the corresponding panchromatic scene and visually inspected at multiple spatial scales. This inspection was conducted using the QGIS software (QGIS Development Team, 2017) and involved inspection of the entire scene as well as a closer look at common problem areas such as along the borders of clouds or where many small icebergs were present. Some scenes were excluded from further analysis on the basis of this quantitative screening and manual inspection. Of the 65 scenes run, ten were excluded at this point, leaving 55 scenes for the six years of our study, with seven to twelve data points per year (mean: 9 acquisition dates per year). Reasons for scene exclusion at this stage included (1) non-detection of large numbers of icebergs greater than one or two pixels in size and (2) large numbers of false positive detections of clouds and/or sea ice as icebergs. Only one scene was removed because of detection failure, due to the presence of mostly transparent clouds, and nine scenes were removed because of false positive detections. Of these nine scenes, seven had values exceeding either the maximum iceberg size or ice–open water ratio thresholds. This inspection of results suggests that although there is no way to automatically screen for scenes with a large number of missed icebergs, the majority of results dominated by false signals can be detected and eliminated automatically through the use of ice coverage and iceberg size cutoffs.

## 2.5 Error Analysis

The primary objective of this investigation was to accurately extract iceberg size distributions using automated methods. To assess inaccuracies in our iceberg size distribution dataset, we evaluate algorithm performance by probing the differences between a manually derived validation dataset and the results produced using the automated approach. To generate a validation dataset, we manually corrected outlines of icebergs in two selected areas in each of four scenes (these validation regions are shown in Fig. 1). These regions were selected to ensure our error analysis included the full spectrum of iceberg size distributions, iceberg concentrations, and cloud cover types and abundances across the four scenes. For each validation region of each scene, a thresholded and vectorized panchromatic image served as a starting point. These outlines were then manually checked and each polygon was either left as is, edited to accurately delineate the iceberg (e.g. where multiple icebergs were combined or, more commonly, portions of the iceberg were missed or represented by multiple polygons), or removed. Icebergs that had been missed by the algorithm were outlined manually and their area was rounded to the nearest pixel (i.e. nearest multiple of $225\ \mathrm{m}^2$) for consistency with the automatically derived datasets. To minimize inconsistencies due to operator bias across the validation datasets, one analyst completed this process for all eight sections.

The primary source of error in these datasets is derived from the digitization process and consists primarily of false positive (i.e. clouds and sea ice identified as icebergs) and false negative (i.e. missed icebergs) detections. The use of an automated method limits bias in determination of iceberg boundaries, and the results are fairly insensitive to the choice of the reflectance



threshold used to identify icebergs in the cloud-masked scenes (Fig. 3 and 4). Specifically, modification of the reflectance threshold by ~5% (~10%) to 0.18 and 0.20 (0.17 and 0.21) resulted in changes in the slope of the power law fits of <2.5% (<4.5%). The observed changes in slope are largely driven by differences in the number of small icebergs identified when the threshold is altered. Large changes in threshold values (e.g. ~50%) tended to result in very large uncertainties (>50-100% for

5   ice coverage and the number of icebergs), and manual inspection of these results showed they were not consistently capturing icebergs (high threshold) or discriminating them from other noise in the image (low threshold).

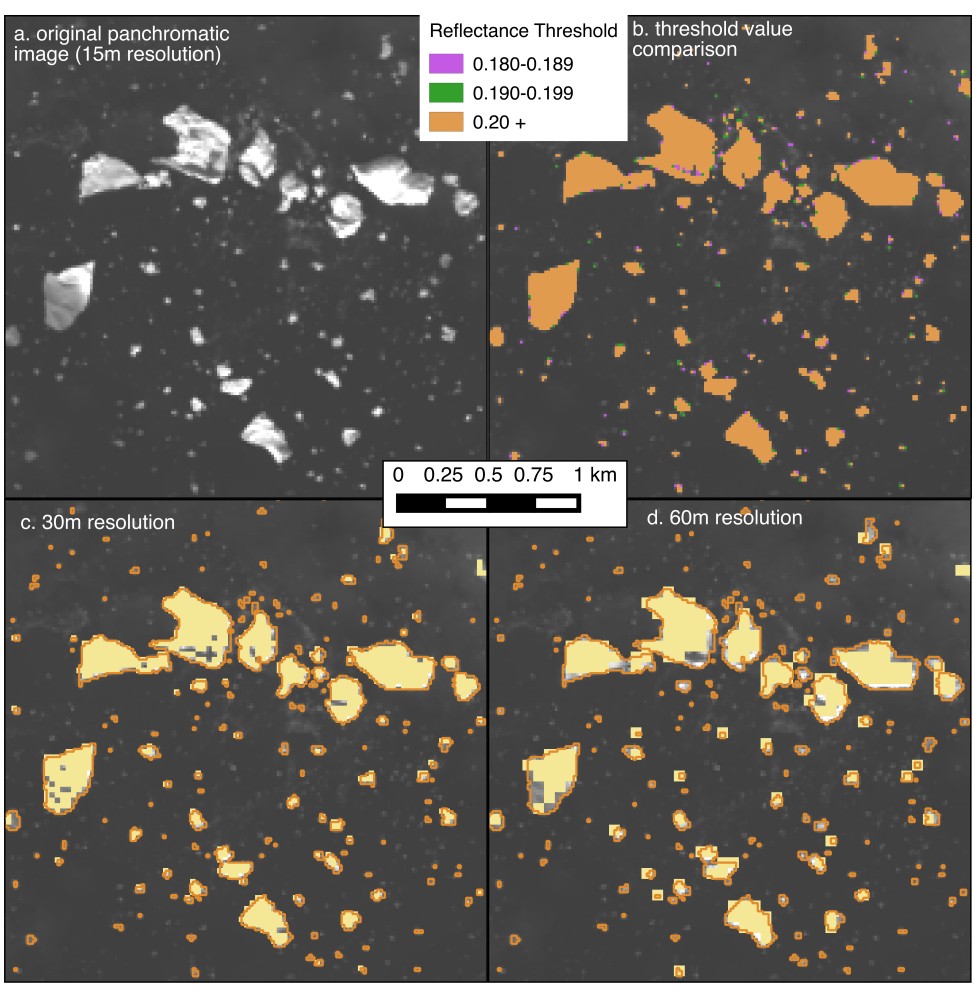

**Figure 3.** Qualitative inspection of threshold choice and image resolution on results. Location of panels is shown on Fig. 5. a) Original panchromatic (Landsat 8 band 8) scene from 31 August 2013. b) The results of threshold sensitivity tests. Pixels identified as ice by the optimal threshold (0.19) are colored orange and green. c–d) Automated iceberg masks constructed with the optimal threshold (in yellow), but for 30 m resolution (c) and 60 m resolution (d) images. Orange lines in (c–d) show iceberg outlines derived from the 15 m resolution image.



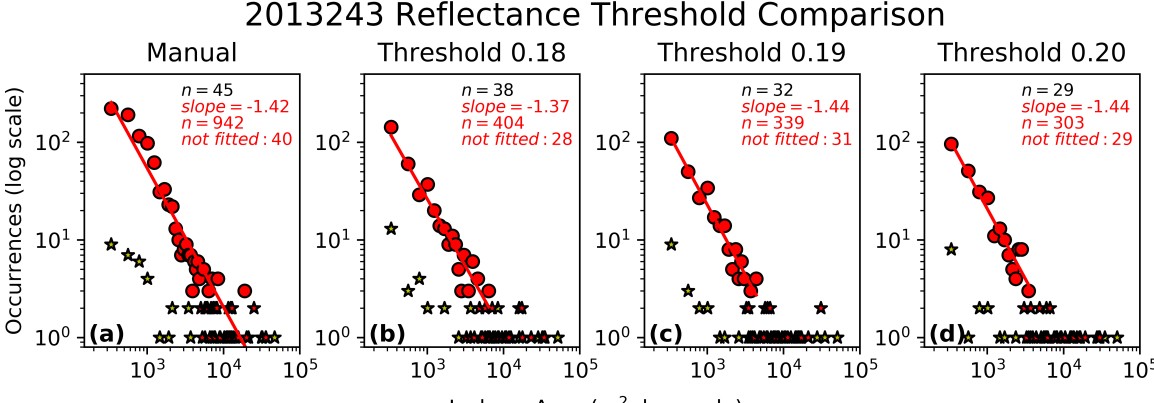

**Figure 4.** Iceberg size distributions derived manually (a) and using several different reflectance thresholds (b–d) for 31 August 2013 (2013243) scene. Iceberg size distributions are shown for the SW (yellow with black text) and mouth (red with red text) regions. Fitted power law curves (by region, in log-log space) show characteristic decay of iceberg areas. Points shown as circles were used to fit the curve; stars show data that was excluded from fitting (not fitted). n is the total number of icebergs detected.

The generation of scene mosaics and upsampling of lower resolution bands also introduces error into our analysis. Since Landsat scenes utilize the UTM projection, the generation of mosaics often requires reprojection of one scene so that the same UTM zone is used by both scenes in the mosaic. During mosaicking, pixel values in overlapping portions of the scenes are averaged. Due to coregistration errors, which can skew the location of icebergs between scenes, pixel averaging has the

potential to lead to both false positive and false negative iceberg identifications: iceberg pixel values may drop below the threshold used for iceberg detection, causing icebergs to be missed, or small icebergs may be double-counted. In line with previous studies, careful qualitative inspection of the overlapping region of several mosaics suggests that the geolocation of the individual Landsat scenes was on the order of the pixel resolution (Storey et al., 2014), which is sufficient for image mosaicking without compromising the detectability of icebergs or resulting in double counting of small icebergs. The impact of upsampling

to match the highest spatial resolution bands is limited to the application of our cloud mask to the panchromatic image. We note that our iceberg size distributions are similar regardless of the amount of cloud cover in a given scene, suggesting that the upsampling of our cloud mask to match the spatial resolution of the panchromatic band does not affect our results.

## 3   Algorithm Outputs

We assessed the validity of our cloud masking and iceberg size distributions using both qualitative and quantitative methods.

The cloud masks were inspected and evaluated qualitatively throughout their generation and testing, while quantitative analysis drove development of the final model used in the machine learning-based cloud mask. A manually derived iceberg validation dataset was used to quantitatively evaluate both the cloud mask and iceberg delineation portions of the algorithm.





### 3.0.1   Cloud Masks

Results from application of our ratio-based cloud masks are shown in Fig. 5. The results produced using the red-SWIR normalized index were similar to those of the individual SWIR reflectance and red:SWIR tests (not shown). The performance of each ratio-based method to distinguish clouds from snow/ice varied widely with cloud type. In particular, very bright, white, fluffy

cumulus clouds, which often had similar reflectance values as icebergs, resulted in abundant false identifications of clouds and/or icebergs. The combination of the NIR:SWIR ratio and SWIR reflectance to identify clouds improved the differentiation between clouds and snow/ice over either method alone and over other combinations of the ratio and threshold methods described above (Fig. 5) and was thus used as a point of comparison for the performance of our machine learning-based cloud mask. However, the ratio-based method suffered from the same problems of its components, making it ill-suited for automated

analysis of a large number of scenes with diverse cloud types. Further, for all ratio-based methods, we could not find a universal threshold that was sufficient for discriminating between clouds and icebergs in all scenes; the threshold value had to be tailored to each scene (Jedlovec, 2009), requiring extensive operator input and inspection.

The edge detection approaches to identify icebergs or clouds also proved inadequate. The feature detection algorithm failed to identify icebergs (or clouds) and was too computationally intensive to run on a laptop, which was one of our design goals to

ensure that the algorithm can be readily used by people outside of the scientific community. Segmentation suffered from similar problems to the ratio-based approaches. In particular, iceberg and cloud objects were segmented simultaneously, providing no distinction between clouds and icebergs (Fig. 5). This result is unsurprising given that the segmentation map was seeded by applying a conservative reflectance threshold to the panchromatic band, which generally fails to differentiate clouds and ice/snow pixels. Manual seeding may have improved results but would require extensive user input for each scene. Random

seeding is also not feasible because it would require dense enough coverage to include a seed in each iceberg and cloud segment (where clouds are present); this would also likely lead to over-segmentation of the image. Given the poor performance of the edge detection methods at distinguishing clouds and ice, they were not pursued further nor compared to the red-SWIR normalized index or machine learning-based cloud masks.

The cloud mask we ultimately selected to include in our algorithm was that derived from the machine learning model.

This cloud mask most consistently and accurately discriminated between bright, white, puffy cumulus clouds and the bright snow/ice surfaces of icebergs without relying on the secondary step for the removal of small groups of pixels that bordered clouds to construct an accurate cloud mask. The ratio-based approaches relied on this secondary step to remove large portions of cloud missed by the mask. As described below, the iceberg size distributions produced using the machine learning-based cloud mask consistently match those of our manual validation dataset, demonstrating its ability to eliminate clouds prior to

iceberg identification. Therefore, we considered the machine learning-based cloud mask to be the best cloud masking procedure moving forward in our analysis.

In addition to the qualitative comparisons of cloud masks illustrated in Fig. 5, we quantitatively assessed the performance of our cloud masks through a comparison of the iceberg size distributions extracted from images after applying the various cloud masks. Specifically, we ran the algorithm and extracted results for each of the two validation regions (described in



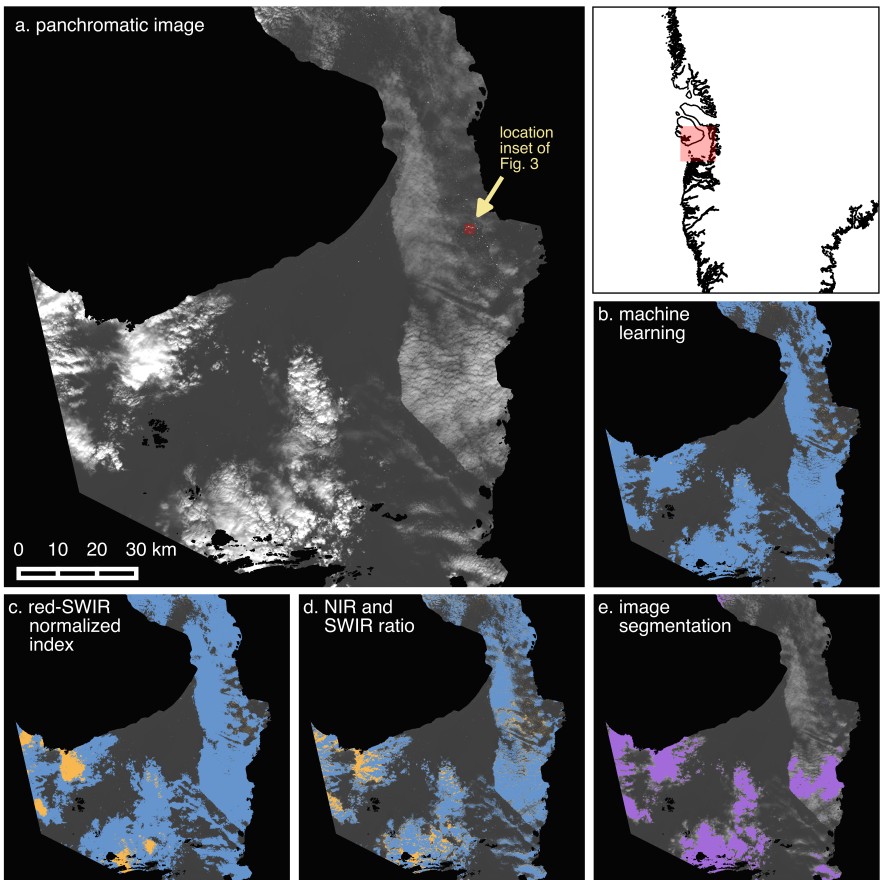

**Figure 5.** Iceberg delineation and cloud masking results using a variety of techniques. a) Panchromatic (Landsat 8 band 8) scene of Disko Bay from 31 August 2013 showing multiple cloud types. b–d) Cloud mask (blue) generated using machine learning, red-SWIR normalized index thresholding, and NIR:SWIR ratio combined with SWIR reflectance band thresholding, respectively. Everything above the iceberg delineation threshold after cloud masking is shown in orange. e) Icebergs and clouds are detected simultaneously using image segmentation (purple).



Sect. 2.5) in four scenes masked with (1) the combination NIR:SWIR and SWIR reflectance ratio-based cloud mask and (2) the machine learning-based cloud mask. We then compared the resultant iceberg size distributions to the manually derived validation dataset. Both methods underestimated the total ice area (median 38% and 32% underestimation for the machine learning- and ratio-based cloud masks, respectively) as well as number of icebergs (median 61% and 57% underestimation).

Further inspection of the extracted iceberg size distributions suggests, however, that the iceberg size distributions extracted following application of the machine learning-based cloud mask are more accurate then those produced following application of the ratio-based cloud mask. Relative to the manually extracted iceberg size distributions, we found that the mean iceberg size was overestimated by 13% (70%) and the slope of the iceberg size distribution was underestimated by 16% (22%), when automatically extracted from machine learning-based cloud-masked (ratio-based cloud-masked) scenes. Manual inspection of

the cloud masks also reveals that the ratio-based mask often classifies significant amounts of open water as clouds, increasing the number of false negative iceberg detections (i.e. missed icebergs) and decreasing the estimated open water area within the ROI.

### 3.0.2 Iceberg Size Distributions

Here we describe iceberg size distributions for each scene using histograms of iceberg areas. Bin sizes increase at one pixel,

or 225 $m^2$, intervals. Previous characterizations of iceberg area size distributions around Antarctica and Greenland have been described using both power law and lognormal distributions (e.g. Tournadre et al., 2012; Enderlin et al., 2016; Kirkham et al., 2017; Sulak et al., 2017). We tested both distributions for scenes containing at least ten bin sizes, excluding bins that contained fewer than three icebergs. Curves were only fitted where ten or more iceberg sizes existed to avoid overfitting in scenes with few icebergs or where extensive cloud cover limited the range of iceberg sizes present. Bins with less than three icebergs

were excluded because they represent the largest iceberg sizes and their distributions are primarily controlled by the maximum iceberg size produced by the infrequent calving of the largest icebergs. Thus, these larger icebergs are poorly captured by both size distribution models. Following Sulak et al. (2017), we selected a power law distribution because it consistently provided a reasonable fit to the data at the scale of the entire bay (Fig. 4). The fitted power law slopes fall within the range −1.22 to −3.09, with a median of −1.76 (Fig. 6), in agreement with previously published values for iceberg size distributions in Greenland's

fjords (−1.9 to −2.3 in Enderlin et al. (2016) and −1.62 to −2 in Sulak et al. (2017)). The fitted power law slopes have a mean uncertainty of 2.5%, calculated as the change in slope value resulting from a ~5% change in threshold value.

  As noted above, the algorithm tends to underestimate both the the total amount of ice area as well as the number of icebergs (by 38% and 61%, respectively). The difference between these numbers suggests that the majority of the missed ice area is made up of icebergs in the smaller size fractions rather than underestimation of the area of larger icebergs. Based on the size

of icebergs commonly added during manual delineations, we define small icebergs as those on the order of 1–5 pixels, or up to 1,125 $m^2$. This idea is bolstered by the observation that the fitted power law slopes tend to be underestimated by a mean of 16%, based on a quantitative comparison between the manually and automatically derived validation datasets. We note that small icebergs that occupy a few pixels or less tend to be missed most often when they are visible through clouds or the density of small icebergs is high. The inability of the algorithm to identify all small icebergs is likely due to mixed pixel effects. Setting





a threshold low enough to detect pixels that are partially ice-filled would result in additional false positive detections of wave crests and other features that may locally enhance the reflectivity of the water surface. We expect that these false positive detections would lead to the considerable overestimation of the number of small icebergs, which would manifest as a very steeply sloped and poorly fit power law curve.

## 4   Example Application and Discussion

Beginning in 1997, Jakobshavn Isbræ underwent a period of rapid retreat, thinning, and acceleration that included the loss of a floating ice tongue in the early 2000s (e.g. Joughin et al., 2004; Luckman and Murray, 2005; Holland et al., 2008; Bondzio et al., 2017). As Jakobshavn transitioned from a persistent floating ice tongue (high stress, easy detachment) to grounded in deeper water (more difficult detachment, easier transport of full thickness icebergs away from the terminus) (Bassis and Jacobs, 2013), the dominant calving style shifted from infrequent calving of tabular icebergs towards more frequent calving and overturning of full thickness icebergs and smaller partial-thickness icebergs (e.g. Amundson et al., 2008; Joughin et al., 2008; Cassotto et al., 2015). These changes in calving style, and consequently calving energies, produce icebergs with different geometries. Since iceberg decay is strongly controlled by iceberg geometry, the change in Jakobshavn's calving style may be reflected as a shift in iceberg size distributions over time.

We applied the fully automated algorithm, using the machine learning-based cloud mask, to the Landsat archive for Disko Bay from 2000–2002 and 2013–2015 (Fig. 6). These year ranges were chosen to span the time period of greatest change in calving behavior of Jakobshavn Isbræ (Amundson et al., 2008; Joughin et al., 2008) as well as avoid missing icebergs in whole or in part due to the striping caused by the SLC failure on Landsat 7. Ice coverage is a function of the total ice-covered and open water areas relative to the observed area for that scene. We present ice coverage as an ice–open water ratio in order to account for differences in scene extent and cloud coverage. Iceberg size distributions are represented by the slope of a fitted power law curve. Maximum detected iceberg size and the number of 225 $m^2$ (one pixel) icebergs delineated in each scene demonstrate the potential influence of many small and/or one large iceberg(s) on the total ice cover in the bay. Some icebergs >2 $km^2$ (approximately an order of magnitude larger than the maximum size of icebergs able to pass over the sill in Ilulissat Isfjord) were not excluded from our results because manual inspection (Sect. 2.4) revealed that they are localized areas of potentially unnavigable sea ice that have little influence on the derived ice–open water ratio.

The iceberg time series constructed for Disko Bay suggests a seasonal pattern in ice cover, with a peak in ice–open water ratio around mid July. The seasonal pattern in the ice–open water ratio observed here for Disko Bay is analogous to that observed by Foga et al. (2014) in Sermilik Fjord in 2008, where the mélange iceberg-to-sea ice ratio showed a seasonal signal with a peak during summer months. We attribute the seasonal pattern in ice–open water ratio primarily to annual variations in iceberg flux, which is dictated primarily by the seasonal presence of sea ice and mélange (so called sikkusak) in the fjord as well as the calving rate of Jakobshavn Isbræ (Joughin et al., 2008; Amundson et al., 2010; Cassotto et al., 2015). In the spring, as sea ice disappears and the mélange becomes less rigid, calving increases (Joughin et al., 2008; Amundson et al., 2010; Cassotto et al., 2015) and icebergs that have become small enough to pass over the sill at the mouth of the fjord are rapidly discharged into the



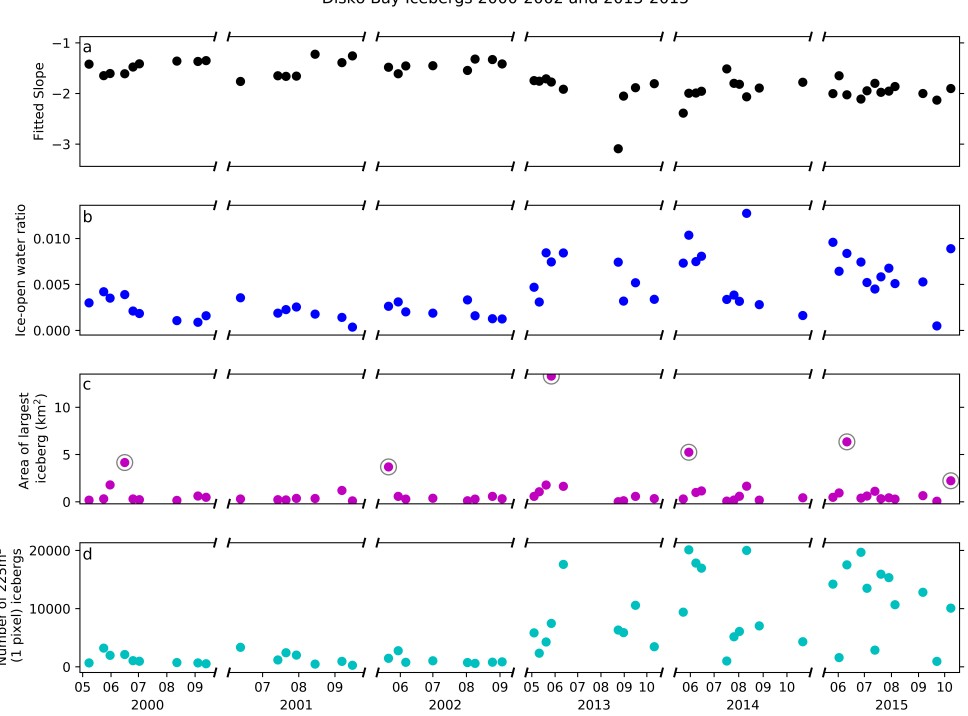

**Figure 6.** Iceberg data extracted from 55 Landsat scenes spanning 2000–2002 and 2013–2015. a) The slope of the power law curve fit to the iceberg size distribution for each scene. Error bars, representing 2.5% of the slope value, are obscured by the symbol. b) Ice–open water ratio. c) Plan view area of the largest iceberg. Circled points are $>2$ km$^2$. d) Number of one pixel (225 m$^2$) icebergs.

bay, causing a spike in ice cover in the bay. As seasonal melting and calving progress, icebergs are free to move into the bay as soon as they can clear the sill until sea ice once again creates a rigid matrix holding them in place and limiting further calving the following winter (Joughin et al., 2008; Amundson et al., 2010; Cassotto et al., 2015).

5   We also observe that the seasonal pattern in ice cover in the bay has evolved over time. During the latter time period (2013–2015), the range of ice–open water ratios is notably larger relative to the turn of the century (2000–2002), concurrent with a distinct increase in the number and frequency of times that a large number of small (one pixel) icebergs are present in the bay. During 2000–2002, none of the scenes analyzed have $>10{,}000$ icebergs in the smallest size bin (225 m$^2$), whereas no less than a fifth of the scenes have $>10{,}000$ small icebergs in 2013–2015 and at least two-thirds of scenes have $>5{,}000$ small icebergs. The increased presence of small icebergs between the 2000–2002 and 2013–2015 observation periods matches anecdotal evidence from boat captains operating in Disko Bay, whom have lamented the difficulty of navigating and setting long fishing lines in recent years due to the large number of small icebergs present in the bay. We attribute this change in the seasonal abundance of small icebergs in the bay to the change in the dominant calving style of Jakobshavn Isbræ. An increase in the number of smaller icebergs can be explained as a consequence of the increase in calving energy, and thus iceberg fragmentation, as Jakobshavn's



terminus geometry evolved from persistently-floating to seasonally-grounded. Additionally, as the glacier terminus geometry evolved, there was likely a decrease in the number of large icebergs stranded on the sill and an increase in the number of small icebergs that could pass over the sill unabated. These changes caused the seasonal pattern in iceberg cover in the bay to become less pronounced as the frequency with which the bay is flooded by small icebergs increased. Although submarine melting of

icebergs will also profoundly affect iceberg size distributions (Kirkham et al., 2017), we hypothesize that temporal changes in submarine melting had little influence on the observed shift in iceberg size distributions for two reasons. First, subsurface ocean observations in Disko Bay suggest that the waters in the bay warmed in the late 1990s and have remained relatively warm since then (Holland et al., 2008; Gladish et al., 2015). Second, the small and presumably shallow-drafted icebergs would have been relatively insensitive to the observed changes in ocean temperatures at depth in the bay.

Interestingly, the size distributions of icebergs in the bay suggests a small but measurable decrease in slope values between the two time periods, concurrent with the changes outlined previously. For 2000-2002, the median slope was $-1.45$ ($\pm 0.03$) while for 2013-2015 the median slope was $-1.92$ ($\pm 0.04$). We note that this pattern is observable despite oceanic warming (Holland et al., 2008; Gladish et al., 2015). Oceanic warming will preferentially decrease the abundance of small icebergs due to their high surface area to volume ratios, with warmer near-surface water temperatures driving increased melt. This will

effectively remove icebergs in the smallest size fractions and will likely manifest in iceberg size distributions as a shift from fitted power law to lognormal size distributions. If we consider that the total volume of melt is a function of both the melt rate and time period, then we can look at changes in iceberg size distributions with distance from the source as an analog for changes in the magnitude of melting. As such, the shift towards a lognormal iceberg size distribution observed by Kirkham et al. (2017) suggests that any melt-driven changes in the size distribution would be counter to what we observe.

## 20  5   Conclusions

Icebergs are a key component of ice–ocean interactions, impacting fjord circulation and stratification, ecosystem structure, freshwater flux, and coastal navigation and infrastructure. Yet even as the flux of icebergs to Greenland's coastal waters has changed in recent decades, constraining the size distribution of icebergs and the controls on these size distributions has been limited to only a few studies. Here we present an automated approach to delineate icebergs in both cloud-free and cloudy

optical (Landsat) satellite images. To test the performance of the automated approach and demonstrate its utility, we provide an example application to six years (2000–2002 and 2013–2015) of Landsat images for Disko Bay, west Greenland.

Differentiation of clouds and snow/ice in optical imagery is challenging given their similar spectral properties and cloud masking often requires a large number of steps and/or advanced computing capabilities. To eliminate clouds from our scenes, we developed a novel and computationally efficient cloud masking scheme using machine learning to identify clouds. A com-

parison of manually extracted iceberg size distributions with those automatically extracted from cloud-masked scenes suggests that the machine learning-based cloud mask performs better than the commonly used ratio-based approaches to delineate snow/ice or clouds. Thus, we recommend that studies leverage similar machine learning-based cloud masking approaches in order to utilize Landsat scenes with partial cloud cover over glaciers and sea ice.




Our iceberg size distribution time series suggests that the ice cover in Disko Bay follows an annual cycle that we attribute to the seasonal presence of sea ice. In the winter months, the sea ice creates a dense mélange that suppresses calving and limits iceberg transport out of the bay. With the spring breakup of sea ice, there is an increase in ice cover on the bay as icebergs are released and calving resumes. This ice cover then decreases throughout the summer/calving season, resulting in a seasonal

signal in ice cover that is in agreement with other investigations. This pattern became less coherent as the size distributions of icebergs in Disko Bay underwent an important transition from the early 2000s (2000–2002) to the early 2010s (2013–2015), concurrent with a transition in Jakobshaven Isbræ's dominant calving style from low energy tabular icebergs to high energy full thickness icebergs. The change in calving style was coincident with a pronounced increase in the number of small icebergs in the bay between 2000–2002 and 2013–2015. The change in abundance of small icebergs led to a decrease in the power law

slope describing the distribution of iceberg sizes and an increase in the total ice coverage in the bay. The temporal change in the number of small icebergs present is in support of anecdotal evidence from local marine navigators and suggests that iceberg size distributions are directly influenced by changes in calving style. This study serves as a starting point for investigating this hypothesis, which will require the application of our algorithm to additional sites to derive iceberg size distribution datasets that span changes in parent glacier calving behavior.

*Author contributions.* JS and GH designed the project and JS carried out the algorithm development and application with feedback from GH and EE. JS prepared the manuscript with contributions from EE.

*Competing interests.* The authors declare that they have no conflict of interest.

*Acknowledgements.* This work was supported by NASA Headquarters under the NASA Earth and Space Science Fellowship Program (grant NNX15AP08H, awarded to JS) and the US National Science Foundation Adaptation to Abrupt Climate Change IGERT Program (grant

DGE-1144423). We thank Jakob Abermann and Eva Mätzler for assisting with project development and insightful discussions on iceberg detection and icebergs along the west coast of Greenland. We also thank the many marine operators and other residents of Nuuk and Ilulissat who provided information on iceberg hazards in Disko Bay and helped shape the goals of this project.





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
