# Peer review of "Automated iceberg detection using Landsat: method and example application in Disko Bay, west Greenland"

_The Cryosphere, 2018_

## Referee Comment (RC1) · Anonymous Referee #1 · 22 May 2018

**1   General comments**

Identifying icebergs in polar imagery is important for estimating the amount of distributed freshwater supply to the fjord, and is an integral component for studying the interaction between ice and ocean. Scheick et al. aim at identifying icebergs in the Disko Bay region in west Greenland. Unlike several similar studies with optical imagery, Scheick et al. do not limit their analysis to cloud-free images, which is potentially a substantial improvement to other studies. They develop a cloud masking technique which facilitates automatic delineation of icebergs just outside the Jakobshavn fjord and into the Disko Bay. Cloud masking is particularly important in optical imagery where the

visible spectral properties of icebergs and sea ice and clouds become similar and improper identification of clouds can lead to erroneous estimates in iceberg area/number in the fjord. The manuscript is well written (although there are confusing sections) and authors lay out the significance and purpose of the work quite well, by explaining some of the already-existing cloud masking literature for optical imagery. They compare their machine learning-based cloud masking technique with the traditional band ratio methods and suggest that their method performs better. In the end they apply their method to find the iceberg size distribution for two time periods between 2000-2002 and 2013-2015. Despite the mentioned strengths of the text, the current scientific content of the manuscript suffers from serious weaknesses.

In brief,

1. The current work fails to fulfil the claims made in the title and abstract: a) it does not correctly detect the icebergs, and b) it is not at all automated. The "real-time" application of the work is also an over-exaggeration, because there are several manual screening steps (to discard inappropriate scenes, etc.) involved in the process.

2. The explanation about the machine learning-based cloud mask is weak, without proper explanation, and lacks clarity.

3. The glaciologic content of the manuscript is very narrow, almost absent (which would make the manuscript more suitable for "methodology" journals).

These are major issues and are grounds for rejection of the current manuscript. Each of the above items is explained in detail in the following section.

Regardless of the limitations of optical imagery for iceberg detection, I think the authors approach is certainly worth pursuing. But I believe that this work requires extensive improvement with greater explanation of the applied techniques. Perhaps, the

current manuscript could be split into two separate units, with the first explaining and fully evaluating the algorithms (both cloud mask and iceberg detection), and the second, application for different fjords/regions in Greenland (to prove that the "automated" algorithms are not region-specific).

**2 Scientific critique**

1. In terms of identifying icebergs, Figure 4 disputes authors conclusion that their algorithm is performing well. In Figure 4, the number of manual picks for the red region is 942, as opposed to 404 for the low threshold (0.18). This is less than 50% of the correct value. I understand that there could be clusters of icebergs determined as one single large iceberg. But this huge underestimation can lead to wrong scientific conclusions (in the absence of manual validation for other regions). Moreover, several iceberg occurrences are unjustifiably dropped from the log-log regression in Figure 4 (denoted by stars), which seems like a cherry-picked analysis.

   Throughout the text, the authors rely heavily on customizing the study region, as well as the images they ended up using (e.g. excluding fjords, sea ice, mélange, etc). In section 2.2 the authors report that they picked May to October to avoid sea ice, and they manually identified images with "such extensive cloud or sea ice cover that icebergs could not be manually identified." In total, from 65 images, 10 were discarded through manual inspection (although authors claim that the cutoff for ice-water ratio detects the majority of them - but that is very region-specific). These examples (and other manual screening steps) make me believe that the algorithm is semi-automated and requires a lot of user interpretation, in addition to its poor performance.

2. The statistical details of the machine learning technique are missing in this paper.
Without seeing any report on the classification score for the classifier method and/or any confusion matrix to demonstrate the suitability of the classifier for this problem, I cannot be convinced that the machine learning algorithm is performing properly. The only instance that performance of the classifier is demonstrated is in Figure 5, which is not sufficient. In the machine learning section, the authors also provide a brief explanation about different edge detection and segmentation algorithms. Those descriptions are vague and inaccurate. Some of them are outlined in the next section of this review.

The authors have used multinomial logit. But there is no explanation why this is a preferred method to the numerous other classification schemes. Multinomial logit, at its core, is a linear regression for discrete classes, so it is certainly not capable of capturing potential nonlinearities between the feature vector and the target variable. It is unclear why authors preferred this type of classification to other classifiers such as random forest, decision trees, SVM, k-NNs, etc. I understand that the computational efficiency is important in this case and neural network classifiers (arguably) may not be used, but random forest classifiers are equally as fast as any linear classifier. I am not claiming that linear regression should not be used, but there must be some explanation/justification as to why this type of regression was chosen. Finally, I wonder how the cloud mask results of this study compares with Landsat's own QA band. There's a range of pixel values for the QA band for difference confidence levels of cloud detection. The comparison with the high-confidence QA band results is also necessary for validation.

3. The implication of this work is limited to a roughly two-page discussion of icebergs distribution in the fjord, which in actuality doesn't say anything new: when Jakobshavn had a floating tongue there were fewer icebergs, but now that it calves the full-thickness ice, icebergs are smaller but more abundant. I acknowledge the fact that this somewhat corroborates the scientific expectation and serves as a "validation tool" for the authors, but for several reasons I contest the conclusions

based on these results: 1) The results presented in the manuscript do not convince me that the algorithm is performing properly; hence, no scientific conclusion shall be drawn based on a dysfunctional algorithm. 2) There is no seasonality. All the results are from May through October which is essentially summer $\pm$ one month.

**3  Minor comments**

P3, Fig. 1. Would be better to use an image that covers the entire Disko Bay region. Also would be useful to mark the location of sill at the fjord mouth.

P3. L7. The terms "false positive" and "false negative" are frequently repeated in the text with brief and repetitive explanations in parenthesis. Perhaps somewhere in the sentence "However, the existing cloud masking approaches..." would be a reasonable place to once and for all define these terms.

P4. L3. Remove "potentially important". There has been detailed explanation in the introduction that these are actually important.

P7. L3. "or classify images into segments (i.e., segmentation)". That is not correct. Segmentation, in essence, does not classify anything. It may require a seed (as mentioned in the last sentence of this paragraph) to identify boundaries of different segments of an image. But classifying what the already-segmented regions represent is a task different from segmentation. Also, some segmentation algorithms require an initial seed (e.g. region-growing or sometimes level-set), but there are many more that don't depend on providing a seed. This paragraph is inaccurate and needs much more work.

P8. Section 2.3.4. In general, what is the need for this ROI? Haven't you mentioned in 2.3.5 that you are ignoring the icebergs that touch/intersect with the polygon shapefile

of Disko Bay region?

P8. L25. "... clusters with a ratio of cloud to pixels greater than 0.4 were reclassified as could edges." Why not 0.3 or 0.5?

P9. Section 2.4. This section is the main part that makes me dispute the automated algorithm, as well as its performance. In L3. Ratio of ice-open water greater than 0.008 is flagged as outlier. Where is that from? If the study site was chosen inside the fjord, then I assume that every single scene would have to be discarded. What is the evidence that this is a reasonable ratio for other regions of Greenland?

P9. L22. "in two selected areas in each of four scenes." I may have missed it, but which four scenes?

P11. Fig. 4. It is confusing to have black text representing the yellow region. Why not making it something consistent like blue? The borders of stars are too thick and obscure the colors. Clearly the threshold cutoff cannot correctly "count" the number of icebergs. But how does the total area of manual compare with the total area of different thresholds? Perhaps (hopefully) those are not this much off.

P14. L6. "... more accurate then those..." replace 'then" with "than".

P14. L17. "We tested both distributions for scenes containing ..." How many more scenes ended up being removed because of this?

P14. L25. "The fitted power law slopes fall within the range ... in agreement with previously published values..." That doesn't seem right. The range of slopes in this paper are -1.22 to -3.09, quite significantly wider range than that of Enderlin 2016 (-1.9 to -2.3) or Sulak 2017 (-1.62 to -2). Considering that this is the slope of a log-log plot, the slopes, of course, are exponents of the relationship between iceberg size and area. Again, this apparent "agreement" is a bit of a stretch.

P15. L8. What is "high stress" here? Deviatoric stresses? Longitudinal stress gradients? Is it strain rates?

P15. L13. "Since iceberg decay is strongly controlled by iceberg geometry..." there needs to be a reference here.

P16. Fig. 6. Because the x axes of this plot vary for every time period, it is hard to follow the "seasonality" of different periods against each other. Also, I contest that these results present any "seasonality". The chosen time frames presented here (May through October) are basically summer $\pm$ a month. So I hesitate to call it seasonal.

---

## Referee Comment (RC2) · Anonymous Referee #2 · 17 Jun 2018

This manuscript aims to tackle an exciting and important problem in current cryospheric science: the automated detection and classification of icebergs from satellite imagery. The topic and scope of the manuscript are a good fit for The Cryosphere. The article is furthermore very well written. However, the manuscript in its present form contains a number of striking weaknesses.

In essence, I fully agree with the comments of Reviewer #1 (and do not believe that a line-by-line review is needed or helpful at this point). The main issues that I believe need to be addressed before this article could be reconsidered for publication are

1) Neither the cloud masking nor the iceberg detection components of the algorithm

are automated. At numerous stages along its execution, an operator has to manually intervene and perform time-consuming tasks (e.g. screen for too much clouds or sea ice, delineate the ROI, exclude regions of iceberg grounding, exclude scenes with too many false positives/negatives).

2) There appears to be substantial tuning of parameters and hand-picked exclusions of problematic zones required in order to make the algorithm work for the given example region. This makes me wonder how readily the algorithm could be applied to different regions. I would argue that such a test should be included in the study to illustrate the usefulness of the method.

3) This may be the main issue: even after substantial manual work, the algorithm does not appear to perform very well. (Which is also an illustration of just how difficult a task this is). The underestimation of the iceberg-covered area by 1/3 and of the number of icebergs by over 50% are, I would argue, unacceptably large errors. One might make the case that the algorithm is more designed to establish the shape of the size distribution (rather than the actual iceberg area/number), but that result is somewhat weak as well: while an underestimation of the slope by 16% might be acceptable, the range of slopes ($-1.22$ to $-3.09$) is much larger than in other published studies. As reviewer #1 discussed in more detail, the description of the machine-learning algorithm and the analysis of its performance are rather thin.

4) Beyond the methods aspect, the novelty of scientific insight is somewhat limited (as also pointed out by Reviewer #1). It would have been interesting to see the full timeseries from 2000-2015, for example. Presumably the algorithm, once set up, is rather fast and should be able to handle greater numbers of Landsat scenes with ease. Granted, there is the issue of SLC failure on Landsat 7, but even after that failure >75% of the data has still been collected and a number of methods have been developed to deal with the data gaps.

---

## Editor Comment (EC1) · G. Catania (Editor) · 18 Jun 2018

Based on the comments from two reviewers and my views as associate editor, I am recommending rejection of this manuscript. I believe that the authors can withdraw the manuscript from their end if they wish to do so. Resubmission of the manuscript to The Cryosphere would be available once the authors have addressed all of the reviewer comments. Since this is likely to take considerable time and effort, I am rejecting the manuscript as it is at this stage.
* * *